# Lipid Signaling Requires ROS Production to Elicit Actin Cytoskeleton Remodeling during Plant Innate Immunity

**DOI:** 10.3390/ijms23052447

**Published:** 2022-02-23

**Authors:** Lingyan Cao, Wenyi Wang, Weiwei Zhang, Christopher J. Staiger

**Affiliations:** 1Department of Biological Sciences, Purdue University, West Lafayette, IN 47907, USA; zhan2190@purdue.edu; 2School of Agriculture and Biology, Shanghai Jiao Tong University, Shanghai 200240, China; 3Department of Botany and Plant Pathology, Purdue University, West Lafayette, IN 47907, USA; plantwang@gmail.com; 4Center for Plant Biology, Purdue University, West Lafayette, IN 47907, USA

**Keywords:** actin cytoskeleton, pattern-triggered immunity (PTI), phospholipase D (PLD), phosphatidic acid (PA), reactive oxygen species (ROS)

## Abstract

In terrestrial plants a basal innate immune system, pattern-triggered immunity (PTI), has evolved to limit infection by diverse microbes. The remodeling of actin cytoskeletal arrays is now recognized as a key hallmark event during the rapid host cellular responses to pathogen attack. Several actin binding proteins have been demonstrated to fine tune the dynamics of actin filaments during this process. However, the upstream signals that stimulate actin remodeling during PTI signaling remain poorly characterized. Two second messengers, reactive oxygen species (ROS) and phosphatidic acid (PA), are elevated following pathogen perception or microbe-associated molecular pattern (MAMP) treatment, and the timing of signaling fluxes roughly correlates with actin cytoskeletal rearrangements. Here, we combined genetic analysis, chemical complementation experiments, and quantitative live-cell imaging experiments to test the role of these second messengers in actin remodeling and to order the signaling events during plant immunity. We demonstrated that PHOSPHOLIPASE Dβ (PLDβ) isoforms are necessary to elicit actin accumulation in response to flg22-associated PTI. Further, bacterial growth experiments and MAMP-induced apoplastic ROS production measurements revealed that PLDβ-generated PA acts upstream of ROS signaling to trigger actin remodeling through inhibition of CAPPING PROTEIN (CP) activity. Collectively, our results provide compelling evidence that PLDβ/PA functions upstream of RBOHD-mediated ROS production to elicit actin rearrangements during the innate immune response in *Arabidopsis*.

## 1. Introduction

Plants lack an adaptive immune response and rely on the machinery of innate immunity to sense and respond to extracellular danger signals, such as microbe-associated molecular patterns (MAMPs) and damage-associated molecular patterns (DAMP) [1]. Recognition of MAMPs or DAMPs by cognate pattern recognition receptors (PRRs) activates basal defense responses, referred to as pattern-triggered immunity (PTI), and protects plants from pathogen infection. Several cognate pairs of MAMP/PRR have been well characterized; for example, bacterial flagellin and elongation factor EF-Tu, or their peptide mimics flg22 and elf26/ac-elf18, are recognized by leucine-rich repeat receptor kinases (LRR-RK), FLAGELLIN-SENSING2 (FLS2) [2] and EF-TU RECEPTOR (EFR) [3], respectively. Chitin, an oligosaccharide of β-1,4-linked GlcNAc and a major component of fungal cell walls, is perceived by the LysM motif receptor kinase LYK5 [4,5]. Further, endogenous secreted polypeptides or molecules released from damaged host cells act as danger signals upon plant infection. Oligogalacturonides (OGs) [6] released from the plant cell wall and AtPEP1 [7], a 23-amino acid endogenous secreted peptide, are detected by wall-associated kinase1 (WAK1) [8] and another LRR-RK PEP1-BINDING PROTEIN1 (PEPR1) [9], respectively. Following the perception of MAMPs or DAMPs, receptor kinase activation and initiation of kinase cascades, a series of short- and long-term signaling events ensue. Hallmark cellular responses that occur within minutes during PTI include: a transient burst of reactive oxygen species (ROS) [10], fluxes of cytosolic calcium [11,12,13], accumulation of the lipid signal, phosphatidic acid (PA) [14,15], as well as remodeling of the actin cytoskeleton [16]. These responses ultimately lead to transcriptional reprogramming, secretion of compounds toxic to microbial invaders, and fortification of the cell wall against attack. In most cases, cause-and-effect relationships between these events remain to be established.

The actin cytoskeleton is a dynamic structure that organizes and supports a plethora of cellular processes including organelle transport, vesicle trafficking, and nuclear positioning. A growing body of research indicates that the actin cytoskeleton is a target for biotic and abiotic stimuli [17,18]. Previously, we established that rapid, transient accumulation of cortical actin filaments is a new hallmark of PTI in *Arabidopsis* epidermal cells responding to pathogenic and non-pathogenic strains of *Pseudomonas syringae* pv. *tomato* (*Pst*) DC3000 as well as diverse danger signals [16,18,19,20,21]. Using reverse genetics and high spatiotemporal resolution microscopy, we established a system to dissect the host cell contribution to actin remodeling and identified two actin-binding proteins as stimulus–response regulators of the cytoskeleton [18]. ACTIN DEPOLYMERIZING FACTOR 4 (ADF4) promotes the severing of actin filaments and facilitates actin disassembly [22], whereas the heterodimeric actin filament CAPPING PROTEIN (CP) binds to filament barbed ends and limits actin assembly from monomers in addition to reducing filament–filament annealing [23,24,25,26].The phenotype of loss-of-function mutants which recapitulate features of MAMP treatments on actin remodeling and single filament dynamics, as well as the observation that *cp* and *adf4* fail to increase actin abundance after elicitation with MAMPs, suggest that both proteins are negatively regulated or inhibited during the innate immune response [19,20]. We proposed a model whereby ADF4 and CP are necessary for actin remodeling during PTI but implicated in different signaling pathways [19,20]. However, the upstream signals necessary to regulate the actin cytoskeletal dynamics during PTI remain to be established.

Phospholipase D (PLD) and its lipid product phosphatidic acid (PA) are implicated in plant immunity [27,28]. *Arabidopsis* has 12 PLD isoforms grouped into 6 classes—PLDα (3), PLDβ (2), PLDγ (3), PLDδ (1), PLDε (1), and PLDζ (2)—based on sequence and substrate requirements [29], and some are induced during the defense response to pathogens [30]. Mutants of *PLDα1* and *PLDδ* are more resistant and susceptible, respectively, to infection by non-host powdery mildews *Erysiphe pisi* or *Blumeria graminis* f. sp. *hordei* [31,32,33]. Further observations show that PLDδ, but not PLDα1, accumulates at plasma membrane sites where cell wall fortification occurs during powdery mildew penetration [31,33]. In addition, the activity and expression of *PLDβ1* is stimulated during pathogen infection [34]. PLDβ1 is the major contributor to PA production during the response of *Arabidopsis* to infection with either necrotrophic *Botrytis cinerea* or virulent *Pst* DC3000, and loss of this isoform results in reduced bacterial proliferation but susceptibility to fungal attack [34]. On the other hand, another study tested a battery of *pld* single, double, and triple mutants and found no difference in bacterial growth with virulent and avirulent strains of *Pst* DC3000 [32]. Collectively, these studies suggest that multiple PLD isoforms participate in defense against microbial invasion, but they do not clearly establish a specific role in PTI or early signaling events.

PLD directly hydrolyzes structural phospholipids to generate a free head group and a second messenger, PA [35]. Previously, we explored the relationship between PA and the actin cytoskeleton in *Arabidopsis* and found that PA binds recombinant CP in vitro, inhibits CP’s ability to bind filament ends, and stimulates enhanced actin polymerization in vivo [24,25]. Because the increase in filament abundance in *cp* mutants resembles actin remodeling induced by MAMP treatment, we tested a role for PLD/PA signaling through CP during the innate immune response. Chemical inhibition of PLD activity or PA production demonstrates that actin remodeling during PTI functions by inhibition of CP activity [20]; however, genetic evidence for PLD function upstream of actin remodeling in defense is lacking, and which isoforms are necessary for PA-induced actin responses remains to be determined. Notably, PLDβ isoforms from both *Arabidopsis* and tobacco have been shown to interact with both G- and F-actin in vitro, and their activity is dependent on the polymerization status of actin [36,37], making them likely candidates to participate in actin remodeling during defense.

ROS produced by the defense-associated NADPH oxidase, RESPIRATORY BURST OXIDASE HOMOLOG D (RBOHD), in *Arabidopsis* is elicited within minutes of MAMP or DAMP treatment [10]. Recently, we found that exogenous ROS treatment triggers actin filament accumulation in leaf epidermal cells and that CP is necessary to transduce RBOHD/ROS signaling to actin remodeling in response to flg22 [21]. Further, our data indicated that flg22-induced ROS generation is strongly elevated when actin dynamics are inhibited. Since PA regulates actin remodeling through CP, we proposed that negative feedback regulation by CP or the actin cytoskeleton exists to modulate ROS production elicited by flg22. However, it remains unclear how PLD/PA signaling is involved and whether PA and ROS signaling cooperate to negatively regulate CP and actin dynamics. It is noteworthy that PLDα1-generated PA functions upstream and binds directly to RBOHD, thereby eliciting ROS production during ABA signaling and stomatal closure in *Arabidopsis* guard cells [38].

Here, we explore the relationship between PA and ROS in the regulation of actin cytoskeleton dynamics during PTI. By combining quantitative fluorescence imaging, reverse-genetic analyses, and chemical complementation approaches, we uncovered a regulatory mechanism whereby both lipid and ROS signaling operate upstream of actin cytoskeleton during PTI. In addition, our data demonstrate that *Arabidopsis* PLDβ isoforms are required for PA-stimulated actin filament accumulation in response to flg22, and that PLDβ-generated PA acts upstream of RBOHD-ROS signaling to induce actin remodeling during the short-term innate immune response. Further, we demonstrate that CP transduces signals from PA and ROS to choreograph the dynamics of actin cytoskeleton during PTI. Finally, our data further support the existence of a feedback loop from CP to regulate flg22-mediated ROS production.

## 2. Results

### 2.1. PLDβ Isoforms Modulate Actin Remodeling in Response to MAMP or DAMP Treatment

To visualize the actin cytoskeleton in plant cells, a reporter comprised of GFP fused with the second actin-binding domain of *Arabidopsis* FIMBRIN1 (GFP-fABD2) [39] was expressed in wild-type Col-0 seedlings. In previous work, we reported that several MAMPs trigger a rapid increase in actin filament abundance in epidermal cells from mature rosette leaves and dark-grown hypocotyls [16,19,20,21]. Because expression of the actin filament reporter in mature rosette leaves varies due to gene silencing, and dark-grown hypocotyls are not ideal for examining pathogen infection, we explored other options. Cotyledons provide a facile system for studying the molecular mechanisms and cellular responses associated with plant-microbe interactions [40,41]. Therefore, we investigated whether actin filament arrays from the epidermal pavement cells of cotyledons respond to MAMP treatment in a similar fashion to rosette leaves and dark-grown hypocotyls. Actin cytoskeleton remodeling in response to short-term (15–20 min) MAMP or DAMP treatment was examined by collecting z-series images from the abaxial epidermal cells of seven-day-old light-grown cotyledons by spinning disk confocal microscopy (SDCM), and actin array organization was analyzed with tools described previously [16,22]. Reminiscent of previous findings, actin filament abundance markedly increased in wild-type epidermal cells following flg22, chitin, OG, and Pep1 treatment, whereas elf18 altered actin accumulation modestly compared with mock treatment (Figure 1A,D). Excluding elf18 treatment, quantitative analyses showed a significant increase in actin filament density after MAMP or DAMP treatment (Figure 1E). In contrast, the extent of filament bundling in cotyledon epidermal cells, as measured with the parameter skewness, remains unchanged after treatment with MAMPs/DAMPs (Figure 1C,F). Collectively, these data indicate that a rapid increase in the abundance of cortical actin filaments following MAMP or DAMP treatment is a general phenomenon conserved in different tissues of *Arabidopsis*, and the cotyledon system is validated to examine actin remodeling during PTI.

Previous studies with chemical inhibitors demonstrated that PLD-generated PA is necessary to elicit actin remodeling during PTI [20]. To test which PLD isoform(s) might be responsible for the PA production implicated in actin cytoskeletal rearrangements during PTI, we marked a collection of *pld* mutants with GFP-fABD2, all of which harbor an exon insertion of T-DNA (Appendix A) and/or have been implicated in response to pathogen attack [31,33,34]. Because flg22 and its cognate PRR, FLS2, are upstream of well-characterized defense signaling pathways, we used flg22 for most further studies. Among the lines investigated, *pldβ1*, *pldβ2*, *pldδ*, and *pld**γ3* failed to respond to flg22 with actin remodeling, whereas *pldα1* and *pldζ2* had a partial or completely normal response (Figure 1A–C and Appendix A). These data indicate that PLDβ1, PLDβ2, PLDδ, and PLDγ3 are required for flg22-triggered actin remodeling during PTI. Because *pldβ1* displayed resistance to bacterial infection by *Pseudomonas syringae pv. tomato (Pst)* DC3000 [34], we chose the PLDβ class for further study and generated a *pldβ1*-*2 pldβ2* double mutant [31] expressing GFP-fABD2 to overcome potential functional redundancy. Analyses by qRT-PCR demonstrated that both *PLDβ1* and *PLDβ2* gene expression was markedly reduced in the *pldβ1*-*2 pldβ2* double mutant (Appendix A). Actin filament remodeling in *pldβ1-2 pldβ2* treated with different MAMPs or DAMPs was examined. Both qualitative and quantitative analyses (Figure 1D–F) showed that actin remodeling in *pldβ1-2 pldβ2* was insensitive to all treatments, indicating that PLDβ is involved in conserved actin rearrangements during PTI. It does not, however, demonstrate the role of PLDβ in response to pathogens or effector-triggered susceptibility. To confirm a role in pathogen response, we infiltrated rosette leaves on mature plants with *Pst* DC3000 and measured bacterial growth at 0, 24, 36, or 48 h post-inoculation (Figure 2A–D). We found that *pldβ1-1*, but not *pldβ2* or *pldβ1-2*, supported significantly less bacterial growth than wild-type rosette leaves, which is consistent with results for the *pldβ1-1* allele from Zhao et al. [34]. Moreover, we observed that a *pldβ1-2 pldβ2* double mutant [31] was significantly less susceptible to virulent *Pst* DC3000, in contrast to the *pldβ1-1 pldβ2* line examined by Johansson et al. [32].

### 2.2. PA Acts Upstream of ROS to Elicit Actin Remodeling during the Innate Immune Response

Both PA and ROS are sufficient to induce actin remodeling when exogenously supplied to wild-type rosette leaf cells, resembling elicitation by flg22, and bypass the need for the respective enzymes, PLD and RBOHD, that generate these hallmark defense signals [20,21]. As a control, the density of actin filaments in the epidermal cells of cotyledon treated with exogenous PA or H_2_O_2_ was measured. The results showed that PA or H_2_O_2_ treatment stimulated an increase in actin filament abundance in a dose-dependent manner (Figure 3B,C and Appendix A), recapitulating the previously reported flg22-induced actin response. Next, we asked whether PLDβ regulates the actin cytoskeletal response during PTI through its product PA. To this end, exogenous PA was supplied to the epidermal cells of *pldβ1-2 pldβ2* seedlings. Quantitative analyses detected a significant increase of actin filaments after PA addition to *pldβ1-2 pldβ2* and following a similar time frame to flg22 treatment (15–20 min; Figure 3B), indicating that PLDβ regulates the actin cytoskeleton through its product, PA. Similarly, we tested whether chemical complementation of *rbohD* using exogenous H_2_O_2_ treatment would recapitulate defense-induced actin remodeling. The results showed that H_2_O_2_ was sufficient to stimulate actin rearrangement in the absence of RBOHD (Figure 3C). Collectively, these findings demonstrate that both PLDβ-generated PA and RBOHD-produced ROS act upstream of actin remodeling during defense signaling.

To address whether there is crosstalk between PA and ROS signaling, chemical complementation experiments were conducted by applying exogenous PA to wild-type or *rbohD* cotyledons, and actin remodeling was evaluated. When compared to mock and negative control (phosphatidylserine or PS) treatment, PA-stimulated cortical actin accumulation was eliminated in *rbohD* (Figure 3B) and demonstrated that RBOHD is required for PA-induced actin cytoskeletal rearrangement. In other words, these results place PLD-generated PA upstream of NADPH oxidase activity and lead to the prediction that exogenous ROS should stimulate actin accumulation in *pldβ1-2 pldβ2*. Consistent with this hypothesis, exogenous H_2_O_2_ overcame the deficiency of PLDβ and triggered an increase of actin density in *pldβ1-2 pldβ2* (Figure 3C). Collectively, we hypothesize that PA acts upstream of ROS signaling to regulate actin rearrangement during PTI.

To test this further, we employed diphenyleneiodonium (DPI), an NADPH oxidase inhibitor [21,42,43]. Pretreatment with DPI significantly reduced flg22-stimulated apoplastic ROS production in leaf disks (Appendix A) and inhibited flg22-induced actin accumulation in the epidermal cells from cotyledons (Figure 4A), resembling the actin cytoskeletal response in *rbohD* mutant (Figure 3A). The requirement of NADPH oxidase activity for actin remodeling in DPI-treated wild-type cells can be bypassed with addition of exogenous H_2_O_2_ (Figure 4C) but not PA (Figure 4B). These results indicate that DPI blocks ROS-mediated actin remodeling elicited by flg22. Next, *pldβ1-2 pldβ2* was pretreated with DPI to block both PLD and RBOHD signaling. If flg22 stimulates actin remodeling by activating PLDβ upstream of ROS production, then exogenous PA treatment should not bypass the requirement for NADPH oxidase activation. We found that the PA-stimulated increase in actin abundance disappeared in *pldβ1-2 pldβ2* treated with DPI (Figure 4B), whereas exogenous H_2_O_2_ triggered an actin cytoskeletal response with or without DPI (Figure 4C).

Similarly, we used the PLD inhibitor, 5-fluoro-2-indolyl des-chlorohalopemide (FIPI) [20,44], to test if PA signaling is upstream of RBOHD. Pretreatment with FIPI inhibited flg22-triggered actin remodeling in cotyledon epidermal cells from wild type (Figure 4D), and significantly reduced apoplastic ROS production in leaf disks (Appendix A), reflective of the scenario in *pldβ1-2 pldβ2* mutant (Figure 3A). Exogenous PA (Figure 4E) or H_2_O_2_ treatment (Figure 4F) could bypass the chemical inhibition of PLD activity in wild type and phenocopy the actin cytoskeletal response in *pldβ1-2 pldβ2* (Figure 3). If PA acts upstream of ROS signaling, genetic disruption of ROS production will be insensitive to upstream signals (such as flg22 or PA), whereas exogenous H_2_O_2_ should bypass this block. In support of this, neither flg22 nor PA triggered actin remodeling in *rbohD*, both with or without FIPI treatments (Figure 4D,E). However, exogenous H_2_O_2_ stimulated actin accumulation in the *rbohD* mutant, in the presence or absence of FIPI, albeit at somewhat attenuated levels compared with mock-treated *rbohD* or wild type (Figure 4F). In sum, these data support the conclusion that PLDβ-generated PA stimulates the production of H_2_O_2_ and leads to actin remodeling during flg22-mediated innate immune signaling.

### 2.3. CP Is a Central Node That Modulates Actin Dynamics in Response to PA and ROS Signaling

Previously, we showed that actin filament capping protein, CP, integrates multiple signals to facilitate cytoskeletal rearrangements during PTI in rosette leaves and hypocotyls [20,21]. To test whether CP functions downstream of PLDβ/PA and RBOHD/ROS signals, we created a *rbohD cpb-1* double mutant expressing GFP-fABD2. Actin remodeling in *rbohD cpb-1*, as well as in *rbohD* and *cpb-1* single mutants, was unresponsive to flg22 treatment when compared to mock-treated wild type (Figure 5A). Exogenous H_2_O_2_ induced an increase in actin filament abundance in both wild-type and *rbohD,* but not in *cpb-1* cotyledon epidermal cells (Figure 5C), similar to rosette leaves [21]. The actin response was ameliorated in *rbohD cpb-1* treated with exogenous H_2_O_2_, similar to the phenotype of *cpb-1* alone (Figure 5C). This provides genetic evidence that ROS functions upstream of CP during actin remodeling. Further, as predicted if PLDβ/PA operates upstream of RBOHD-ROS, the actin cytoskeletal response was abrogated when exogenous PA was supplied to *rbohD cpb-1* (Figure 5B). Consistent with CP serving as a modulator of actin remodeling that functions downstream of PLD/PA and RBOHD/ROS signals, *cpb-1* alone or treated with DPI or FIPI failed to respond to flg22, PA, or H_2_O_2_ (Figure 4B–F and Figure 5A–F). This was also tested genetically by creating a *pldβ1-2 pldβ2 cpb-1* triple mutant. As expected, the triple mutant failed to remodel actin in response to flg22, PA, or H_2_O_2_ treatment (Figure 5D–F). Taken together, our data demonstrate that CP is a central node to coordinate actin cytoskeletal rearrangements during MAMP-triggered immunity in *Arabidopsis*.

### 2.4. PA Regulates flg22-Triggered ROS Production to Mediate Actin Rearrangements

From the data above, we conclude that PA acts upstream of RBOHD to modulate actin remodeling in response to flg22 treatment. To examine whether PA regulates RBOHD-generated ROS production during PTI, we employed the rosette leaf system, which is widely used to measure apoplastic ROS after MAMP treatment [40]. Specifically, ROS produced by leaf disks of wild type and *pldβ* mutants following flg22 treatment was measured. Apoplastic ROS production was stimulated within minutes of MAMP treatment and peaked at around 12 min in wild-type leaf disks, whereas the response was completely abrogated in *rbohD* leaf disks (Appendix A), as shown previously [21]. Since all data related to flg22-stimulated actin response were obtained in the cotyledon system, we performed similar measurements of apoplastic ROS production in cotyledons. The results verify the timing and extent of ROS response in wild-type and *rbohD* cotyledons (Appendix A) and demonstrate that RBOHD is required for ROS production in both rosette leaves and cotyledons. For ease of sample preparation and abundance of material, we used the rosette leaf system to further test our hypothesis about the mechanism of ROS production. To statistically compare ROS production among different genotypes, we analyzed the data at the peak of 12 min. Apoplastic ROS production in *pldβ1-1, pldβ2,* and *pldβ1-2 pldβ2* was significantly reduced, whereas there was a modest but not significant reduction of ROS detected in *pldβ1-2* compared to wild type treated with flg22 (Figure 6A and Appendix A). Moreover, combining loss of RBOHD with mutants for PLDβ by creating *pldβ1-2 rbohD*, *pldβ2 rbohD*, and *pldβ1-2 pldβ2 rbohD* double and triple mutants completely abrogated flg22-elicited ROS production (Figure 6B and Appendix A). These data provide the first evidence that PLDβ contributes to flg22-triggered apoplastic ROS production in *Arabidopsis*.

Previously, we showed that the actin cytoskeleton and/or CP mediates negative feedback regulation of ROS production [21]. In *cpb-1*, flg22-mediated ROS production was significantly elevated when compared to wild type, and this response requires RBOHD because the *cpb-1 rbohD* double mutant fails to generate ROS (Figure 6C and Appendix A). We hypothesize that this CP–actin negative feedback loop requires the PLDβ/PA signaling pathway. To test this, we examined ROS production in *pldβ1-2 pldβ2 cpb-1* and *pldβ1-1 cpb-1* following flg22 treatment. If the elevated ROS in *cpb-1* is attributed to feedback stimulation of PA level in cells, removal of PLDβ in the signaling pathway would limit PA accumulation, which would limit ROS levels elicited by flg22. However, the data show that flg22-induced ROS production in *pldβ1-2 pldβ2 cpb-1* and *pldβ1-1 cpb-1* mutant was much higher than wild type or *cpb-1* single mutant (Figure 6B,C and Appendix A), which is opposite to the prediction. The data do, however, seem to indicate the presence of a potential feedback loop from actin cytoskeleton to ROS production that operates via PLDβ, which is more complicated than our previous model.

### 2.5. CP and RBOHD Are Epistatic to PLDβ during Bacterial Pathogen Proliferation

To further explore the relationship between PLDβ, RBOHD, and CP, growth of virulent *Pst* DC3000 in inoculated rosette leaves from four-week-old plants was measured (Figure 2). As noted above, growth of DC3000 was significantly reduced in *pldβ1-1* and *pldβ1-2 pldβ2* compared to wild type, but not in *pldβ1-2* or *pldβ2* single mutants (Figure 2C,D). These results are consistent with PLDβ playing a negative role in the battle against bacterial invasion. On the other hand, as reported previously [17], CP plays a positive role in defense against DC3000, and *cpb-1* supports more bacterial growth than wild type (Figure 2C–E). Interestingly, we found that plants became more susceptible than or showed no difference from wild type plants when CP was disrupted in the *pldβ1-1* and *pldβ1-2 pldβ2* background (Figure 2B–D). These results indicate that PLDβ and CP function in the same pathway during microbial defense and that CP acts downstream in the process. Similarly, we found that the *pldβ1-2 pldβ2 rbohD* triple mutant was more susceptible to bacterial infection compared to *pldβ1-2 pldβ2*, indicating that RBOHD is epistatic to PLDβ and providing additional evidence for them acting in the same pathway (Figure 2D). Moreover, the *rbohD cpb-1* double mutant was more susceptible to bacterial infection than WT seedlings or the respective single mutants, indicating that RBOHD and CP are both positive regulators of defense. These results support the hypothesis that PLDβ acts upstream of RBOHD during the defense against bacterial infection (Figure 6D).

## 3. Discussion

In this study, we proposed a hypothetical working model that places PLDβ/PA upstream of RBOHD/ROS production in the regulation of actin dynamics during plant immune response (Figure 6D). When *Arabidopsis* cells are challenged with a pathogenic microbe such as *Pst* DC3000, the pattern-recognition receptor (PRR), FLS2, senses the danger signal by recognizing the immunogenic peptide flg22 and activates the PRR complex partners, BAK1 and BIK1. Immune response activation of PLDβ occurs through an unknown mechanism, but could involve direct phosphorylation by the PRR kinase complex or by stimulation from the cytosolic Ca^2+^ flux binding to the C2 domain of PLDβ [27,28]. Production of PA by PLDβ targets RBOHD to enhance the burst of apoplastic ROS during defense. The signaling molecule ROS, in turn, inhibits the activity of CP and thereby enhances actin filament assembly and overall dynamicity leading to remodeling and increased abundance of filaments. Actin filament dynamics and/or CP feeds back to upstream steps of the pathway, including the reduction of ROS production, as shown previously [21], or earlier by modulating PLDβ activity, as suggested here. Our model illustrates that CP is a convergence point for multiple signaling pathways to modulate actin cytoskeletal dynamics during the response to microbial invasion.

### 3.1. The Actin Cytoskeleton Is a General Target during Plant–Microbe Interactions

Dynamic remodeling of the actin cytoskeleton is critical for a wide spectrum of cell activities across diverse organisms. During microbial invasion of plant tissues, actin cytoskeletal organization changes rapidly to accommodate the subcellular reallocation of different components for defense [17,18,45,46,47,48,49]. Over the same time scale, several other cellular events occur, such as ROS burst [10], cytosolic calcium fluxes [11,12,13], and PA accumulation [14,15]. This study and our previous work [20,21,25] demonstrate that actin cytoskeletal remodeling is downstream of both ROS and PA during the early PTI signaling pathway. Those data highlight the executive role of actin cytoskeleton network in response to pathogen infection.

Previously, we found that the abundance of cortical actin filaments increased significantly in epidermal cells of rosette leaves and dark-grown hypocotyls of *Arabidopsis* in response to MAMP or DAMP treatment [16,19,20,21]. Here, we broadened the survey to include cotyledon epidermal cells and observed a similar actin filament accumulation following PTI signals. However, actin filament remodeling was unresponsive to the immunogenic peptide, elf18, similar to previous observations on rosette leaves [16,21]. It is possible that the elf18-EFR-triggered actin cytoskeletal response only occurs in certain cell types and tissues, such as the dark-grown hypocotyl [19,20], or that elf18-EFR-related actin response is inhibited in light-grown tissues. Collectively, these studies highlight the fact that rapid remodeling of actin array organization is a conserved response or hallmark event during PTI present in epidermal cells from diverse plant organs.

A key question in this field is why host-cell actin filaments assemble quickly following perception of microbial invaders. One proposed model is that a rearranged actin cytoskeletal network provides tracks for the delivery of defense materials; this is supported by observations of an actin patch as well as actin cables focusing on the penetration site during pathogenic and non-pathogenic fungal and oomycete interactions [46,50,51,52]. These actin-based structures facilitate the deposition of callose, which fortifies the plant cell wall and helps abrogate pathogen invasion [19,20,53]. Successful immunity in plant cells requires intact machinery of PRRs, whose precise localization is critical for PTI. Upon activation by flg22, FLS2 internalizes into endosomes observed in the cytoplasm. However, disruption of actin filament networks with latrunculin B (LatB) perturbs internalization of FLS2 [54], indicating that the actin cytoskeleton is pivotal for intracellular traffic of FLS2 [41,54]. The abundance and activity of other defense proteins at the PM may also be regulated by actin dynamics. It is reported that turnover of RBOHD depends on clathrin-mediated vesicle trafficking in *Arabidopsis* [55]. Live-cell imaging indicates that the enzymatic activity of GFP-RBOHD is promoted by flg22 treatment [55]. Previously, we demonstrated that LatB treatment and CP knockdown mutants produce excess apoplastic ROS triggered by flg22 [21], revealing that a functional actin cytoskeleton is critical to maintain moderate ROS generation during pathogen invasion. One possibility is that cortical actin dynamics facilitate the turnover of RBOHD by endocytosis following immune signaling. Alternatively, similar to the immune receptor, AtHIR1 [56], cortical actin filaments may govern the diffusion, formation of immune complexes, or protein density of RBOHD in the PM. All of these arguments point out that the actin cytoskeleton has broad impacts on intracellular events for successful defense during plant–pathogen interactions.

The actin cytoskeletal network of plant cells undergoes constant rearrangements, even in the absence of extracellular signals [57,58]. Normal actin organization is conferred primarily by the dynamic behavior of single actin filaments and choreographed through a plethora of actin-binding proteins with specific activities [17,59]. Using a powerful combination of high spatiotemporal resolution imaging and genetic tools, we and others have demonstrated that turnover of single actin filaments is precisely regulated through nucleation, severing, and availability of actin filament plus ends [22,25,58,60,61]. The accumulation of actin filaments during PTI could be contributed through increased actin nucleation by the Arp2/3 complex or formins [52,62] or through inhibition of filament severing by ADF4 [19]. Heterodimeric capping protein (CP) limits actin filament assembly by preventing addition of monomers to plus ends and preventing filament–filament annealing [24,25] and has been implicated in multiple signaling pathways [20,21,25,37]. In the *cpb-1* knockdown mutant, it was originally observed that more actin filament ends are available, which recapitulates the response of actin filaments to treatment with PA or MAMPs [20]. Those free filament ends allow for actin subunit addition and facilitate the annealing of actin filaments, thereby contributing to the increase in actin filament abundance. Loss of CP increases overall actin filament array dynamicity, whereas CP overexpression suppresses overall dynamicity, measured by correlation coefficient analysis [26]. This is consistent with the report of a biphasic decrease in actin array dynamicity following MAMP (elf26 and chitin) treatments [20]. Thus, it is hypothesized that CP activity is inhibited by these signaling pathways. Through generation of double and triple mutants, we demonstrated that CP is necessary for both PLDβ/PA- and RBOHD/ROS-directed actin cytoskeletal remodeling in response to flg22 treatment. In addition, *cp* mutants are more susceptible to fungal and bacterial infection than are wild-type plants [20], reinforcing the importance of CP in defense against pathogen invasion. Altogether, we propose that CP is a central node to coordinate multiple signaling pathways that regulate actin cytoskeleton dynamics.

### 3.2. PLDβ/PA Requires RBOHD to Modulate Actin Cytoskeletal Remodeling during PTI

During plant–microbe interaction, several intracellular events, including production of PA and a burst of apoplastic ROS, occur roughly coincidentally with actin cytoskeleton remodeling. In this study, we demonstrated that both PA and ROS act upstream to stimulate the actin cytoskeletal rearrangement in response to flg22 and that RBOHD is necessary for PA-induced actin rearrangement during PTI (Figure 3). Further evidence from pharmacological approaches, including H_2_O_2_-stimulated actin rearrangement in the *rbohD* mutant with FIPI treatment (Figure 4F) and lack of PA-induced actin rearrangement in *pldβ1-2 pldβ2* mutant after DPI treatment (Figure 4B), indicates that PA is upstream of RBOHD-ROS signaling. The mutual interaction between PA and ROS has long been recognized in the literature. Exogenous PA triggers the generation of ROS in both *Arabidopsis* and rice cells [63], whereas H_2_O_2_ treatment activates PLD enzyme activity resulting in PA accumulation [64]. However, this study provides the first evidence linking PA to ROS signaling in the modulation of actin dynamics during defense against bacterial phytopathogens. The next question is how exactly PA regulates the ROS signaling pathway during pathogen infection. AtRBOHD/F has been identified as a target of PLDα-generated PA to regulate the generation of ROS during the ABA response [38]. It has been well characterized that the mutations of PA-binding motifs in RBOHD abolish ABA-stimulated ROS production through regulating its enzymatic activity [38]. In this study, flg22-triggered ROS production markedly decreased in *pldβ* double mutants, which totally depend on the availability of RBOHD (Figure 6 and Appendix A). Therefore, we propose that PA probably regulates the enzymatic activity of RBOHD during bacterial defense.

The relationship between the PLDβ/PA, ROS, and actin remodeling pathway described here and the immune response against bacterial infection is complicated and perhaps indirect. For example, whereas flg22-elicited apoplastic ROS production is suppressed in the *pldβ* single and double mutants (Figure 6), Zhao et al. [34] demonstrated that ROS is hyper-elevated in *pldβ1-1* following infection with *Pst* DC3000. It is important to remember that actin remodeling is biphasic during virulent bacterial infection of *Arabidopsis* rosette leaves, with an early increase in actin filament abundance linked to PTI and a subsequent reduction in overall actin filament abundance and increased extent of bundling resulting from the delivery of bacterial effectors during effector-triggered susceptibility [16]. Janda and coworkers provided evidence that actin disassembly causes elevated SA levels and that this may be linked to increased plant resistance [65,66]. Similarly, PLDβ1 is a negative regulator of SA biosynthesis and signaling during *Pst* infection but a positive regulator of JA-related defense gene expression [34]. Consequently, translating links between the signal-mediated actin remodeling during PTI into understanding resistance against bacterial invasion should be pursued with caution.

### 3.3. Actin Cytoskeleton Feedback Regulates ROS Production during PTI

Previously, we proposed that elevated ROS production in *cpb-1*, compared to wild type, is mediated through negative feedback regulation of RBOHD in the flg22-FLS2 signaling pathway [21]. Based on our new data placing PLDβ/PA upstream of ROS production, we speculate that loss of CP results in elevated PA accumulation on the plasma membrane, which augments RBOHD-dependent ROS generation by flg22 (Figure 6D). This model should be tested directly, and the recent characterization of a ratiometric PA biosensor, PAleon, holds great potential for future work [67]. Here, the data implicate that PLDβ is indeed required for the feedback regulation of flg22-induced ROS generation. However, instead of inhibiting the ROS burst, we observed much higher ROS production in *pldβ1-2 pldβ2 cpb-1* after elicitation with flg22 compared to either wild-type plants or the *cpb-1* single mutant (Figure 6C). This potential negative feedback loop could operate at the level of PLD/PA, or even further upstream during PTI signaling. A direct interaction between plant PLDβ and both monomeric and filamentous actin has been demonstrated previously [68]; however, this seems an unlikely candidate for the negative feedback mechanism here since PLD activity is stimulated by actin filaments and inhibited by monomer binding. That would correspond to a positive feedback loop from CP and/or actin dynamics during PTI, which is not consistent with the ROS response observed.

The generation of ROS is not solely dependent on response to biotic and abiotic stimuli. However, importantly, ROS as a signal mediator triggers intracellular signaling cascades in living organisms. Vascular plants generate precise mechanisms to maintain the proper concentration of ROS to avoid injury from excess ROS generation. During plant–microbe interactions, BIK1, a component of the FLS2 PRR complex, phosphorylates RBOHD to promote apoplastic ROS generation in *Arabidopsis* [69,70]. Further, PA binds directly to the N-terminus of RBOHD and positively regulates ROS production during ABA-mediated stomatal closure [38]. Finally, activation of RBOHD may require calcium signals [69,70]. However, the above regulatory mechanisms are positive regulators of RBOHD-mediated ROS production. Results from our previous report [21] as well as this study show that the actin cytoskeleton, or CP itself, is a negative regulator of ROS production. After treatment of wild type with LatB or in *cpb-1*, flg22-triggered ROS production is significantly promoted [21], which hints at potential negative feedback regulation by actin filaments or CP to modulate flg22-induced ROS generation. Similar observations were made with *prf3* mutants, disrupted for a novel isoform of the G-actin binding protein AtPROFILIN 3, in *Arabidopsis* leaf disks treated with flg22 [71]. It is possible that the regulation of RBOHD trafficking, membrane dynamics, or immune complex formation by the actin cytoskeleton are critical to balance ROS generation in plant cells. Other negative regulators of ROS production include DYNAMIN-RELATED PROTEIN 1A and 2B (DRP1A and DRP2B) that are implicated in clathrin-mediated endocytosis of FLS2 [41,72]. Combining these reports with our data, it is possible that regulation of RBOHD vesicular trafficking is an important mechanism to attenuate ROS signaling in response to biotic and abiotic stimuli. A detailed dissection of the correlation between RBOHD dynamics and actin cytoskeleton in response to MAMP or DAMP treatment would be useful to test these hypotheses.

## 4. Materials and Methods

### 4.1. Plant Material and Growth Conditions

*Arabidopsis thaliana* ecotype Columbia-0 (Col-0) was used in this study. T-DNA insertion lines for *pldα1* (SALK_053785), *pldβ1-1* (SALK_079133), *pldβ1-2* (SALK_004324), *pldβ2* (SALK_113607), *pldγ3* (SALK_084335), *pldζ2* (SALK_094369), *cpb-1* (SALK_014783), *rbohD* [10], *pldδ* (SALK_023247), and *pldδ-2* (SALK_092469) were obtained from the *Arabidopsis* Biological Resource Center (ABRC; Ohio State University, Columbus, OH, USA) or generous colleagues as described in the acknowledgements. For actin cytoskeleton analyses, Col-0 transformed with GFP-fABD2 was crossed with T-DNA insertion lines, homozygous mutants and wild-type siblings expressing GFP-fABD2 were recovered from F2 populations, and F3 plants were used for data collection. For double and triple mutants, a binary vector with GFP-fABD2 was introduced using the agrobacterium-mediated floral dipping method and selected on plates containing the appropriate antibiotic. T3 seedlings were used for data collection and analyses. For experiments on cotyledons, seeds were sown on ½-strength Murashige and Skoog medium (MS, Sigma-Aldrich, St. Louis, MO, USA) plates solidified with 1% agar, stratified at 4 °C for 2 days, and then grown vertically under long-day conditions (16 h light, 8 h dark) at 21 °C for 7 days. For leaf disk collection, seeds were sown directly in soil and grown under long-day conditions at 21 °C for 4 weeks.

### 4.2. Image Acquisition and Quantitative Analysis of Actin Organization

All seedlings for cotyledon experiments were floated on sterilized distilled water for 16–24 h before any treatments and/or imaging in order to minimize stimulation resulting from wounding. To observe actin filaments in cotyledons, an Olympus IX-83 inverted microscope equipped with a spinning disc confocal head (Yokogawa CSUX1-A1, Hamamatsu Photonics, Hamamatsu, Japan) and an Andor iXon Ultra 897BV EMCCD camera (Andor Technology, Concord, MA, USA) were utilized. All images were collected with an Olympus 100x oil objective (1.45 NA UPlanSApo; Olympus America, Inc., Waltham, MA, USA). To visualize GFP signal, cotyledons were excited with 488 nm light and 15 consecutive images of 12.5 μm total depth were captured with a 0.5 μm z-step interval and emission wavelength at 525–530 nm using MetaMorph version 7.8.8.0 software. A Gaussian blur filter was applied to the images for each z-stack, which was converted to a maximum intensity projection, followed by high-band pass filtration prior to analysis of percentage of occupancy or density [16]. All experiments were repeated at least three times independently to draw conclusions.

### 4.3. Plant Treatments with MAMPs, DAMPs or Chemical Inhibitors

Immunogenic MAMPs/DAMPs, including flg22 (1 μM), Pep1 (1 μM; NeoBioSci, Cambridge, MA, USA), chitin (1 μM; Sigma-Aldrich), or OG (50 μg μL^−1^; courtesy of Bruce Kohorn, Bowdoin College, Brunswick, ME, USA), were diluted in sterile ddH_2_O to prepare a 10X stock, as reported previously [20]. Instead of elf26, as used previously [19,20], here we used the stronger MAMP peptide, ac-elf18 [73]. Whole seedlings were incubated in MAMP or DAMP solution for 20 min, and cotyledons were excised from seedlings prior to imaging. To inhibit general PLD activity, 5-fluoro-2-indoyl des-chlorohalopemide (FIPI; Santa Cruz Biotechnology, Dallas, TX, USA) was used as described previously [20]. In other experiments, the NADPH oxidase inhibitors diphenyleneiodonium or DPI were used to block ROS production [21]. Solutions with peptide or chemical inhibitors were updated every 2 h to ensure full activity. For cotreatments, seedlings were pretreated for 10 min with 50 μM FIPI or 50 μM DPI and then transferred to solution containing inhibitor plus flg22 for an additional 20 min before imaging.

### 4.4. Apoplastic ROS Measurement

Leaf disks of 0.5 cm diameter were punched from 4-week-old rosette leaves and floated in 96-well plates for 16–24 h in advance of elicitation [21,40]. For the test of ROS production in cotyledons, cotyledons were detached from the 8-d-old seedlings and floated in 96-well plates for 16–24 h in advance of elicitation. Prior to ROS measurement, 100 μL solution including 34 mg/mL luminol (Santa Cruz Biotechnology), 20 mg/mL horseradish peroxidase (Sigma-Aldrich), and 1 μM flg22 was replaced for each well in the plate and then immediately transferred to a luminescence plate reader (BioTek Synergy 2 system, Agilent, Santa Clara, CA, USA). For FIPI or DPI treatments, 200 μL solution at the specified concentration was added, and leaf disks were pretreated for 30 min before ROS measurements. All data for direct comparison of treatments and genotypes were collected on the same plate.

### 4.5. Disease Assays

To measure bacterial growth in *Arabidopsis* wild-type and mutant lines, 4-week-old plants were hand-infiltrated with *Pst* DC3000 suspension at 1 × 10^5^ colony-forming units (CFU) with 0.025% Silwet L-77 using a needleless syringe, and the excess solution and associated bacteria on the leaf surface were carefully removed [17]. Plants were covered with a clear plastic tray and harvested at 0, 24, 36, or 48 h after inoculation. Leaf disks were harvested from infected rosette leaves using a hole punch (0.5 cm diameter) and ground in 10 mM MgCl_2_. Following bacterial recovery, serial dilution of leaf extracts was performed; 5 µL from each dilution was placed onto NYGA (1.5% Bacto agar, 0.5% Bacto peptone, 0.3% yeast extract, 2% glycerol, and pH adjusted to 7.0 with 0.5 M NaOH) plates, and the bacteria were grown at 28 °C for 1–3 days for bacterial colony counts [20,73].

### 4.6. RNA Extraction and qRT-PCR

Total RNA was extracted from 7-d-old light-grown seedlings. The total of 50 whole seedlings per genotype were transferred to liquid nitrogen immediately after harvest and ground into powder with precooled pestles. TRIzol reagent (Invitrogen, Carlsbad, CA, USA) was used for RNA purification according to the manufacturer’s protocol. A total of 2 μg of RNA was reverse transcribed using SuperScript II Reverse Transcriptase (Invitrogen) and Random Primer Mix (New England Biolabs, Ipswich, MA, USA). To quantify the *PLDβ1* and *PLDβ2* expression levels in the cotyledons, qRT-PCR was performed using FastStart SYBR Green master mix and a LightCycler 96 thermocycler (Roche, Indianapolis, IN, USA). Primers for *PLDβ1* and *PLDβ2* are shown in Appendix A. For all experiments, gene expression was normalized to *UBC* transcript levels. Three biological and technical repeats were performed for each genotype.

### 4.7. Accession Numbers

Sequence data from this article can be found in GenBank/EMBL data libraries under accession numbers At5g47910 (*RBOHD*), At3g15730 (*PLDα1*), At2g42010 (*PLDβ1*), At4g00240 (*PLDβ2*), At4g35790 (*PLDδ*), At4g11840 (*PLD**γ3*), At3g05630 (*PLDζ2*), and At1g71790 (*CPB*).

## Figures and Tables

**Figure 1 ijms-23-02447-f001:**
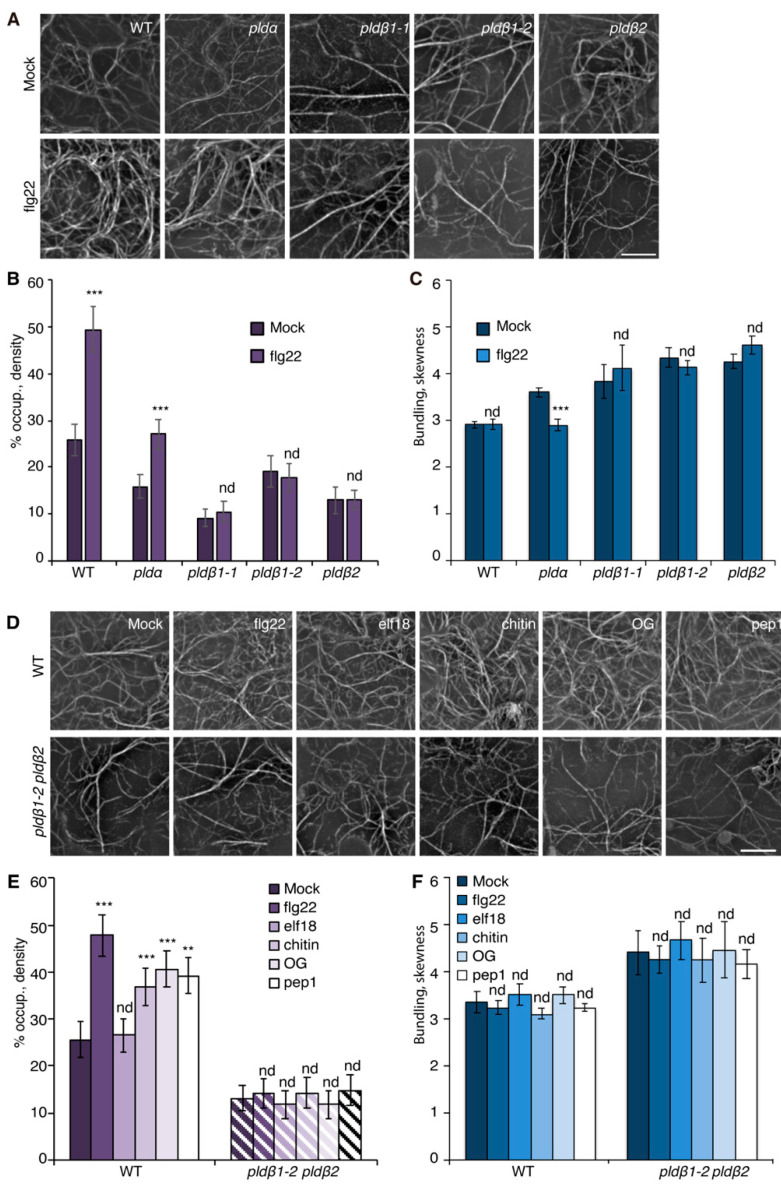
PLDβ is required for actin remodeling in response to various MAMPs/DAMPs. (**A**) Representative images of actin filaments in epidermal cells of cotyledons from wild-type (WT), *pldα1*, *pldβ1-1*, *pldβ1-2*, and *pldβ2* cotyledons. Seedlings were treated with mock or 1 µM flg22 for 15 min prior to imaging by spinning disk confocal microscopy (SDCM). In WT cells, actin filament abundance increased after flg22 treatment. Similarly, actin filaments in *pldα1* cells became more abundant after flg22 treatment. However, there was no obvious alteration to actin filament arrays in *pldβ* mutants. Bar = 10 μm. (**B**) Quantification of actin filament abundance or density in WT and *pld* mutants. Compared to mock treatment, the density of actin filaments was significantly increased in both WT and *pldα1* cells following elicitation with flg22. However, no significant changes were elicited by flg22 treatment for any *pldβ* single mutant. (**C**) The extent of filament bundling (skewness) was measured for WT and *pld* mutants with mock or flg22 treatment. Compared to mock, the skewness values in *pldβ1-1*, *pldβ1-2*, and *pldβ2* cotyledons were not altered after flg22 treatment. However, actin filaments in *pldα* became slightly less bundled after flg22 treatment. (**D**) Representative images of actin filaments in epidermal pavement cells of WT and *pldβ1-2 pldβ2* double mutant seedlings that were treated with various MAMPs or DAMPs for 20 min. Bar = 10 μm. (**E**) Quantitative analysis of actin filament density. In WT, compared to mock, the percentage of occupancy, or density, of actin filaments increased significantly after 20 min treatment with 1 μM flg22, 1 μM chitin, 50 μg μL^−1^ OG, or 1 μM pep1. However, the density of actin filaments after 1 µM elf18 treatment showed no significant change compared to mock-treated WT. The density of actin filaments in epidermal cells from *pldβ1-2 pldβ2* changed slightly but showed no significant differences between mock and MAMP/DAMP treatments. (**F**) The extent of filament bundling (skewness) was measured in epidermal cells of WT and *pldβ1-2 pldβ2* seedlings after various elicitor treatments. The data implied that actin filament bundling was insensitive to those treatments in both WT and *pldβ1-2 pldβ2* mutant. Values given represent mean ± SE; *n* = 50 images from 10 seedlings per treatment and genotype; ** *p* < 0.01, *** *p* < 0.001; nd, no difference, Student *t*-test.

**Figure 2 ijms-23-02447-f002:**
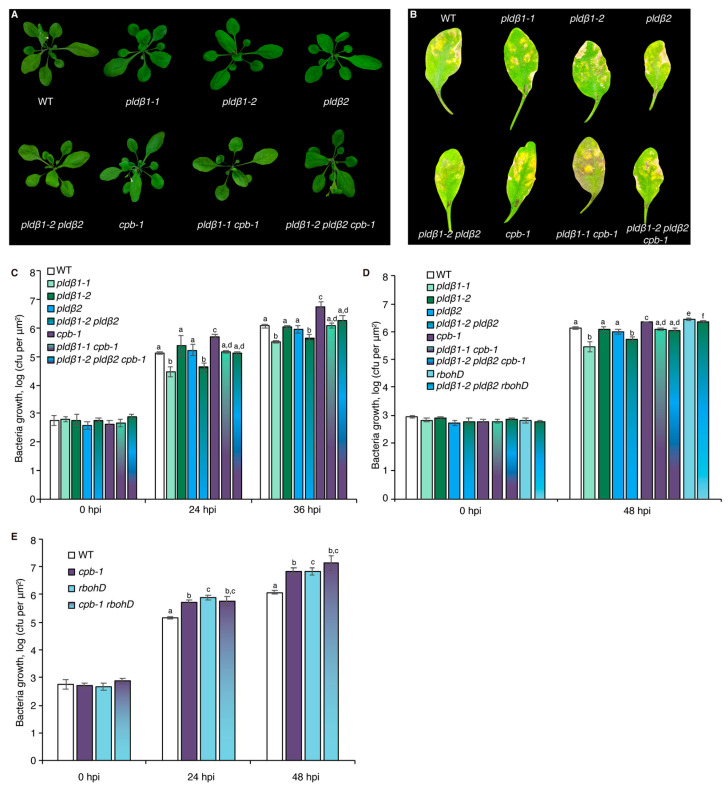
PLDβ acts upstream of RBOHD to defend against bacterial pathogen proliferation. (**A**) Representative images of *A. thaliana* plants including WT (Col-0) and *pldβ1-1*, *pldβ1-2*, *pldβ2*, and *cpb-1* homozygous single mutants, *pldβ1-1 cpb-1* and *pldβ1-2 pldβ2* double mutants, as well as *pldβ1-2 pldβ2 cpb-1* triple mutant. Plants were cultivated for 4 weeks under long-day conditions and photographed. No obvious phenotypic differences were detected in *pldβ* single, double, or triple mutants compared to WT. (**B**) Qualitative images of infected leaves from WT, *pldβ1-1*, *pldβ1-2*, *pldβ2*, *cpb-1*, *pldβ1-1 cpb-1, pldβ1-2 pldβ2* as well as *pldβ1-2 pldβ2 cpb-1* triple mutant at 36 h post infection (hpi). (**C**) Quantification of bacterial growth in *Arabidopsis* leaves infiltrated with *Pseudomonas syringae* pv. *tomato* (*Pst*) DC3000. Leaves were sampled at 0, 24, and 36 hpi to measure the growth of bacterial pathogens. At 0 hpi, there were no significant differences for bacterial growth detected among the genotypes tested. At 36 hpi, *pldβ1-1* (light green) and *pldβ1-2 pldβ2* (gradient from dark green to blue) supported significantly less bacterial growth compared to WT (white), whereas bacterial growth in *pldβ1-2* (dark green) and *pldβ2* (blue) was indistinguishable from WT; *cpb-1* (purple) was susceptible to bacterial pathogen infection compared to WT, supporting more growth of virulent *Pst* DC3000, as reported previously [20]. Moreover, when *cpb-1* was combined with *pldβ1-1* and *pldβ1-2 pldβ2* mutants, the double (gradient from light green to purple) and triple (blend of dark green, blue and purple) mutants were more susceptible to bacterial growth than *pldβ1-1* and *pldβ1-2 pldβ2* alone. This epistatic genetic relationship indicated that CP functions downstream from PLDβ. At least three independent biological repeats with similar findings were conducted. (**D**) Another independent bacterial infection assay was performed with the mutants in (**C**), as well as *rbohD* (light blue) and *pldβ1-2 pldβ2 rbohD* (gradient from light blue to green). Similar conclusions about *pldβ* mutants and *cpb-1* could be drawn from this repeat. Here, *rbohD* was susceptible to virulent pathogen growth at 48 hpi, as expected. Moreover, *pldβ1-2 pldβ2 rbohD* was susceptible to bacterial infection compared to WT and opposite to the resistance against bacterial growth shown by *pldβ1-2 pldβ2* alone. This epistatic genetic relationship indicated that RBOHD functions downstream of PLDβ in the battle against bacterial pathogen invasion. (**E**) Further examination of bacterial growth in *cpb-1 rbohD* double mutant indicated that both CP and RBOHD are positive regulators during pathogen infection. Values represent mean ± SE; different letters indicate statistically significant differences at the same time point, one way ANOVA, compared with Turkey–Kramer honestly significant difference (HSD) in Microsoft Excel.

**Figure 3 ijms-23-02447-f003:**
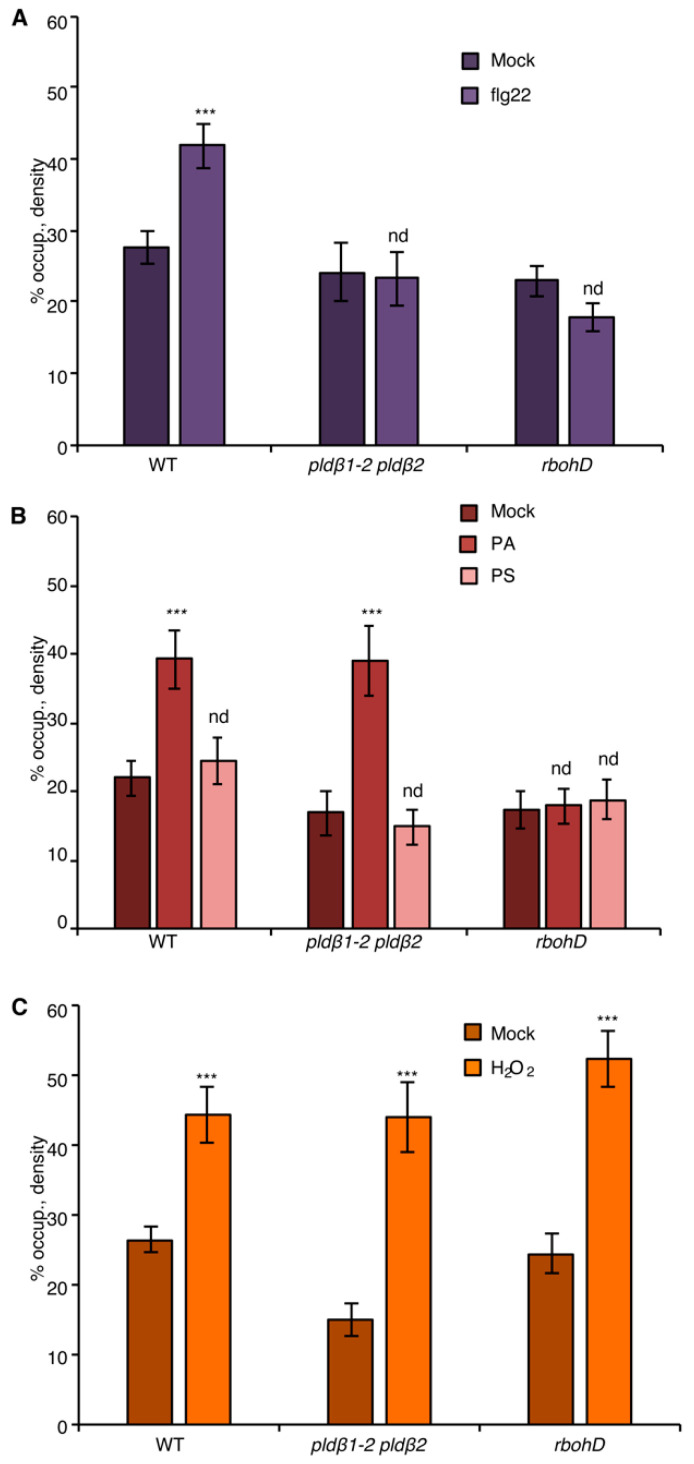
The NADPH oxidase, RBOHD, is necessary for PA-mediated actin remodeling. (**A**) Actin filament abundance in epidermal cells of WT, *pldβ1-2 pldβ2*, and *rbohD* cotyledons after mock or flg22 treatment. Compared to mock treatment, the density of actin filaments elicited by flg22 increased significantly in WT, but not in *pldβ1-2 pldβ2* or *rbohD* mutants. (**B**) Actin filament abundance in WT, *pldβ1-2 pldβ2*, and *rbohD* seedlings after treatment with 50 μM exogenous phosphatidic acid (PA) for 20 min. In WT, compared to mock, PA stimulated an increase in actin filament density. Similarly, the density of actin filaments increased significantly in *pldβ1-2 pldβ2* mutant after chemical complementation with PA. By contrast, there was no significant change in actin filament density in the *rbohD* mutant after treatment. Phosphatidylserine (PS; 50 µM) was used as a negative control for the same experiments. (**C**) Actin filament density in WT, *pldβ1-2 pldβ2*, and *rbohD* seedlings after 1 μM H_2_O_2_ treatment for 20 min. In WT, compared to mock, H_2_O_2_ induced an increase in actin filament density. Similar increases in actin filament density were also measured in *pldβ1-2 pldβ2* and *rbohD* mutants when treated with exogenous H_2_O_2_. Values given represent mean ± SE; *n* = 50 images from 10 cotyledons per genotype and treatment; *** *p* < 0.001; nd, no difference, Student *t*-test.

**Figure 4 ijms-23-02447-f004:**
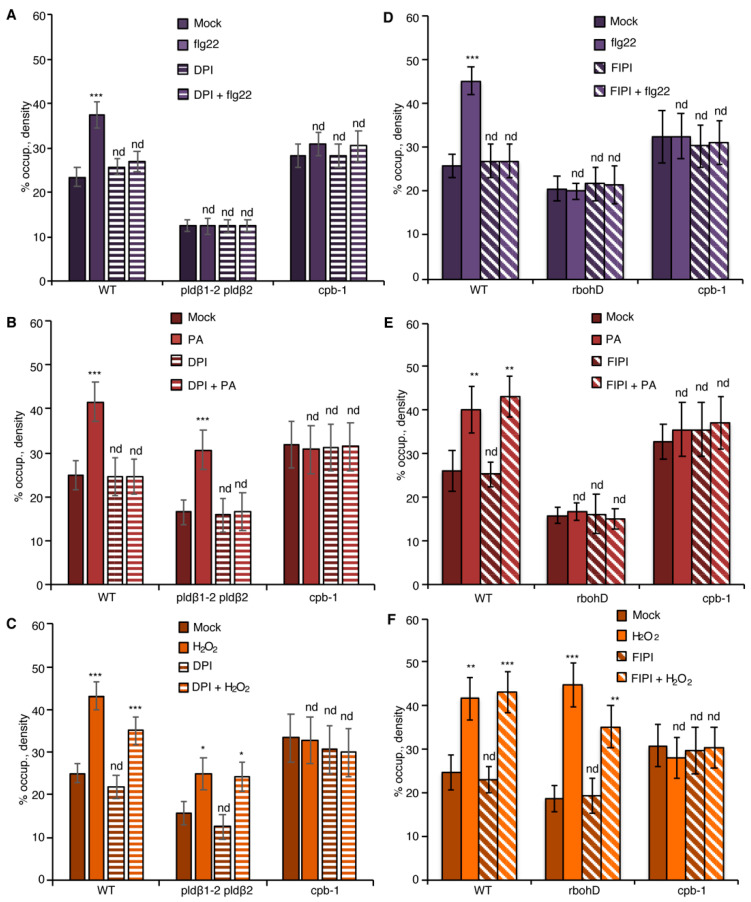
PA regulates actin dynamics through ROS production during PTI. (**A**) Actin filament abundance in WT, *pldβ1-2 pldβ2*, and *cpb-1* seedlings after treatment with flg22 and/or the NADPH oxidase inhibitor, DPI. In WT, flg22-stimulated actin accumulation was impaired in the presence of 50 µM DPI. However, actin filaments in *pldβ1-2 pldβ2* and *cpb-1* were insensitive to any treatment. (**B**) Actin filament abundance in WT, *pldβ1-2 pldβ2* and *cpb-1* seedlings after treatment with PA and/or DPI. PA treatment led to an increase in density of actin filaments in both WT and *pldβ1-2 pldβ2* mutant compared to mock. The presence of DPI blocked PA-triggered increases in actin filament density. In contrast, actin filament organization in *cpb-1* mutant was unresponsive to either treatment. (**C**) Actin filament abundance in WT, *pldβ1-2 pldβ2*, and *cpb-1* seedlings after treatment with H_2_O_2_ and/or DPI. Treatment with H_2_O_2_ induced an increase in actin filament density, either in the presence or absence of DPI, in both WT and *pldβ1-2 pldβ2* compared to mock. However, in *cpb-1*, actin filaments failed to respond to any treatment. (**D**) Actin filament abundance in WT, *rbohD* and *cpb-1* seedlings after treatment with flg22 and/or the phospholipase D inhibitor, 5-fluoro-2-indolyl des-chlorohalopemide (FIPI). The increase of actin filament density elicited by flg22 was inhibited when WT cells were pretreated with 50 µM FIPI for 20 min. By contrast, in *rbohD* and *cpb-1* mutants, actin filaments failed to respond to either treatment. (**E**) Actin filament abundance in WT, *rbohD*, and *cpb-1* seedlings after treatment with PA and/or FIPI. In WT, compared to mock, the density of actin filaments increased after treatment with PA. Even after pretreatment with FIPI, PA still triggered an increase in actin filament density. By contrast, no actin remodeling was detected after either treatment in *rbohD* and *cpb-1* mutants. (**F**) Actin filament abundance in WT, *rbohD*, and *cpb-1* seedlings after treatment with H_2_O_2_ and/or FIPI. H_2_O_2_ induced an increase in actin filament density, in either the presence or absence of FIPI, in both WT and *rbohD* compared to mock. In contrast, actin filaments in *cpb-1* failed to respond to any treatment. Values given represent mean ± SE; *n* = 50 images from 10 seedlings per genotype and treatment; nd, no difference, * *p* < 0.05, ** *p* < 0.01, *** *p* < 0.001, Student *t*-test.

**Figure 5 ijms-23-02447-f005:**
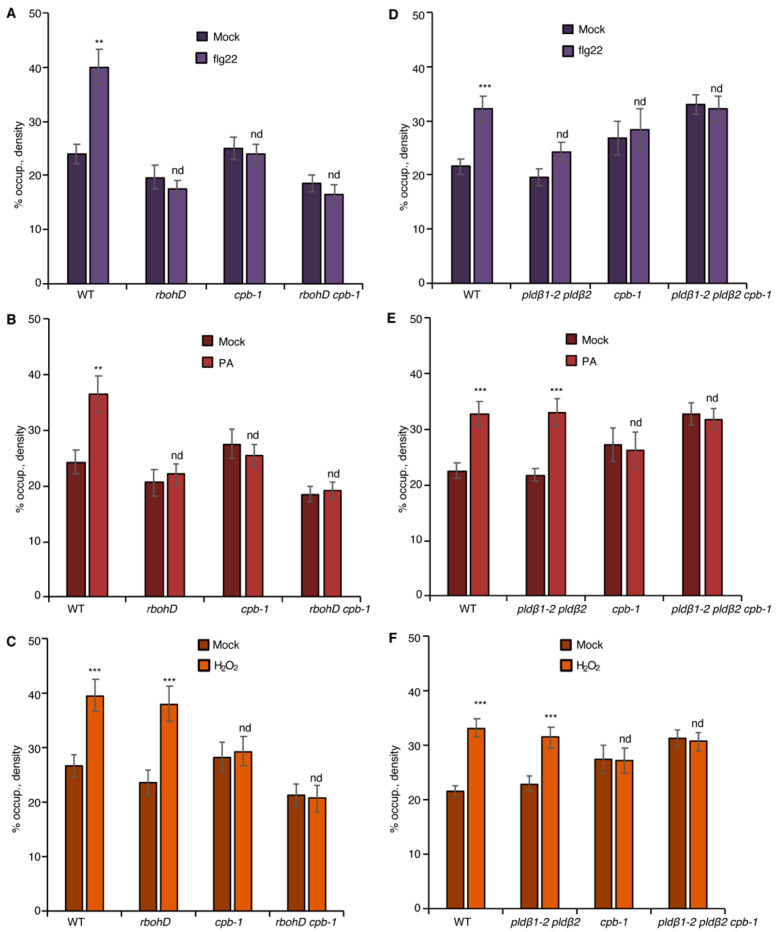
CP transduces both PLDβ/PA and RBOHD/ROS signals to regulate actin remodeling. (**A**) Analysis of actin filament abundance in WT, *rbohD*, *cpb-1*, and *rbohD cpb-1* seedlings after flg22 treatment for 20 min. Remodeling of actin arrays in *rbohD*, *cpb-1*, and *rbohD cpb-1* epidermal cells was insensitive to flg22 treatment. (**B**) Analysis of actin filament abundance in WT, *rbohD*, *cpb-1,* and *rbohD cpb-1* seedlings after treatment with 50 μM PA for 30 min. Actin filament abundance was significantly increased by exogenous PA treatment in WT but not in *rbohD*, *cpb-1*, or *rbohD cpb-1* mutants. (**C**) Analysis of actin filament density in WT, *rbohD*, *cpb-1*, and *rbohD cpb-1* seedlings after treatment with 1 μM H_2_O_2_ for 20 min. H_2_O_2_ triggered significant actin accumulation in *rbohD* as well as in WT seedlings. In contrast, actin filament remodeling in *cpb-1* and *rbohD cpb-1* mutants was unresponsive to H_2_O_2_ treatment. (**D**) Analysis of actin filament abundance in WT, *pldβ1-2 pldβ2*, *cpb-1*, and *pldβ1-2 pldβ2 cpb-1* after flg22 treatment for 20 min. Actin filaments in *pldβ1-2 pldβ2*, *cpb-1*, and *pldβ1-2 pldβ2 cpb-1* failed to rearrange in response to flg22. (**E**) Analysis of actin filament abundance in WT, *pldβ1-2 pldβ2*, *cpb-1*, and *pldβ1-2 pldβ2 cpb-1* after treatment of 50 μM PA for 30 min. Actin filaments in *pldβ1-2 pldβ2* accumulated after PA stimulation, whereas actin filaments in *cpb-1* and *pldβ1-2 pldβ2 cpb-1* were unresponsive. (**F**) Analysis of actin filament abundance in WT, *pldβ1-2 pldβ2*, *cpb-1*, and *pldβ1-2 pldβ2 cpb-1* after 1 μM H_2_O_2_ treatment for 20 min. Similar to WT, actin filaments in *pldβ1-2 pldβ2* increased in response to H_2_O_2_ treatment. However, *cpb-1* and *pldβ1-2 pldβ2 cpb-1* were insensitive to the treatment. Values given represent mean ± SE; *n* = 50 images from 10 seedlings per genotype and treatment; nd, no difference, ** *p* < 0.01, *** *p* < 0.001, Student *t*-test.

**Figure 6 ijms-23-02447-f006:**
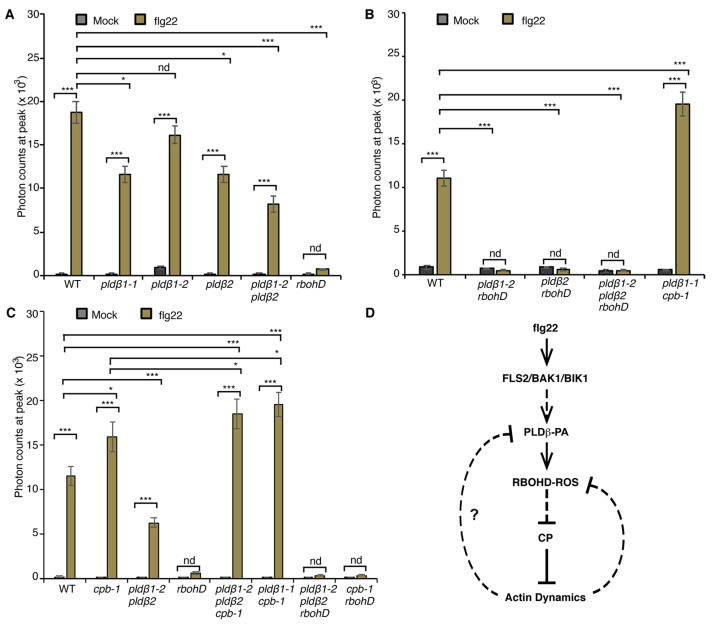
PLDβ is implicated in flg22-triggered ROS production. (**A**) Apoplastic ROS production was measured in leaf disks from 4-week-old WT, *rbohD*, *pldβ1-1*, *pldβ1-2*, *pldβ2*, and *pldβ1-2 pldβ2* mutants. ROS production was detected after flg22 treatment in all genotypes tested, except for *rbohD*. When compared to WT, flg22-induced ROS production in *pldβ2* and *pldβ1-2 pldβ2* was significantly reduced, whereas it changed only slightly in *pldβ1-2*. In *rbohD*, ROS production elicited by flg22 was inhibited. Values given represent mean ± SE; *n* = 16 leaf disks per genotype. (**B**) Apoplastic ROS in WT, *pldβ1-2 rbohD*, *pldβ2 rbohD*, *pldβ1-2 pldβ2 rbohD*, and *pldβ1-1 cpb1-1* mutants. ROS production was inhibited in all mutant combinations containing *rbohD*. (**C**) Apoplastic ROS in WT, *cpb-1*, *pldβ1-2 pldβ2*, *rbohD*, *pldβ1-2 pldβ2 cpb-1*, *pldβ1-2 pldβ2 rbohD*, and *rbohD cpb-1* mutants. Compared to mock, flg22-induced ROS burst in WT was abolished in all the *rbohD* mutant backgrounds. By contrast, ROS production in *cpb-1* and *pldβ1-2 pldβ2 cpb-*1 increased significantly, whereas in *pldβ1-2 pldβ2* it was decreased compared to WT; nd, no difference, * *p* < 0.05, *** *p* < 0.001, Student *t*-test. (**D**) A simple model for the crosstalk between PA signaling, ROS signaling and actin dynamics during flg22/FLS2-mediated immune response. Upon perception of flg22 by its cognate receptor FLS2, PA signaling is activated and then boosts the production of ROS by RBOHD. Both PA and ROS signaling target CP to regulate actin dynamics during plant innate immune response. A potential feedback pathway from actin cytoskeleton would in turn modulate PA and ROS signaling (solid lines: direct interaction; dashed line: potential interaction; arrows: activation of signaling cascades; bars: inhibitory effect).

## Data Availability

The data supporting the findings of this study are available from the corresponding authors, Lingyan Cao and Christopher J. Staiger, upon request.

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
