# Peer review of "Lipid Signaling Requires ROS Production to Elicit Actin Cytoskeleton Remodeling during Plant Innate Immunity"

_ijms, 2022, doi:10.3390/ijms23052447_

Round 1

Reviewer 1 Report

This manuscript by Cao et al addresses the upstream signals that simulate remodeling of the actin cytoskeleton observed during plant immune response. Remodeling of the actin cytoskeleton, along with elevated reactive oxygen species (ROS) production and accumulation of the lipid signal phosphatidic acid (PA), are key hallmarks of cellular response that occurs within minutes during pattern-triggered immunity (PTI). However, the cause-and effect relationships among these events have yet to be established. In this study, the authors investigated the relationship between PA, ROS, and regulation of actin cytoskeleton dynamics during PTI, using fluorescence imaging, reverse-genetic analysis, and chemical complementation approaches.

     Overall, the experiments and data in the manuscript and supplemental material are presented in a straight-forward manner and are of suitable quality. This work has potential significance in that PLD-beta/PA function upstream of RBOHD-mediated ROS production to elicit actin rearrangement during PTI in Arabidopsis. However, it may require more detailed discussion and/or evidence to further validate the mechanisms for the actin cytoskeleton remodeling associated with ROS and PA. Addressing the following points could improve the manuscript.

  • In line 517, the authors mentioned that “normal actin organization is conferred primarily by the dynamic behavior of single actin filaments.” The assay used in the study does not quantify single filaments, but rather quantifies the density of the entire actin cytoskeleton. Is the observed increase in actin cytoskeleton density due to increase in filament number, filament length, filament thickness, or perhaps bundle formation? Further, it would be useful if authors can add detailed discussion on how the intensity of actin network can be related to the actin cytoskeleton remodeling dynamics.
  • Throughout this research article the authors postulate that CP negatively regulates actin assembly (lines 66-67; 354, 523). Although CP does bind to the barbed ends of filaments, it is incorrect to term this “negative” actin regulation. While CP does limit individual filament elongation, on a larger scale CP is required for Arp2/3 complex-mediated actin network assembly. CP regulation of actin is neither positive nor negative, as actin assembly is a dynamic process. It would be more correct to note “CP modulates (or regulates) actin assembly.” To use the term “negatively regulates” it would have to be specifically “CP negatively regulates actin filament elongation.” To say that “CP negatively regulates the dynamics of actin filaments (line 523) would mean that dynamics (both assembly and disassembly) are decreased, which is not the case.
  • Recent publication in Nature (https://www.nature.com/articles/s41467-021-25682-5) demonstrates that CP couples two opposed processes (nucleation and capping) in branched actin network assembly. What is the implication for increased branching for plant immunity? Is there a way to evaluate the extent of branching before and after PTI?

Reviewer 2 Report

In this manuscript, Cao et al. investigated the role of phosphatidic acid (PA) and reactive oxygen species (ROS) in the regulation of actin cytoskeleton dynamics.  After validating that Arabidopsis cotyledons exhibit increased actin filament abundance/density after MAMP treatment, the authors screened a series of phospholipase D (PLD) knockouts and showed that pldβ1, pldβ2, pldδ and pld?3 mutants are impaired for flg22-induced actin remodeling. A pldβ1-pldβ2 double knockout is marginally (0.2 - 0.3 logs) less susceptible to Pto DC3000 than WT plants.  Genetic dissection of the signaling events downstream of PLD indicated that PA-induced actin remodeling is dependent on the NADPH oxidase RBOHD and is regulated by the actin filament capping protein CP.  Furthermore, PLD is a significant positive regulator of flg22-induced ROS production, while CP appears to have a negative regulatory function.  Overall, this manuscript features a selection of genetic and chemical perturbations that are very nicely deployed to convincingly characterize the signaling pathway from PLD-derived PA through to RBOHD-produced ROS.  The downstream events and the ultimate connection between actin remodeling and immunity are less clear, but presumably remain a target for future work.  The manuscript would benefit from minor revisions, as outlined below.

The primary revision required for this manuscript relates to the order of presented data.  Specifically, the data in all figures should appear in the order in which it is described in the Results section.  For example, data for the pldb double mutant in Figure 1 are described after the single mutant data in Figure 2.  The single mutant data should appear before the double mutant data, perhaps in the same figure.  Likewise, the cpb-l data in Figures 3 and 5 are not described until later in the manuscript, after Figure 6 has been referenced.  Here, either the images should be re-arranged among the figures to match their order of appearance in the Results section, or the Results section should be revised to match the figures, perhaps by presenting a compare-and-contrast of WT/pldb/cpb genotypes within each relevant section of the Results.  Finally, sections 2.2 and 2.3 should be combined to provide a more cohesive story. 

Much of the narrative of this manuscript is that the elicitation of PTI is associated with increased actin filament abundance, which in turn should positively impact the immune response.  This was borne out by previous results with the disruption of the actin network by latrunculin B, which resulted in enhanced susceptibility to P. syringae.  Here, however, a pldβ1-pldβ2 double knockout with reduced MAMP-responsive actin phenotypes and diminished ROS production is less susceptible to infection.  In contrast, cpb-1 mutants have more abundant actin filaments and produce more ROS than WT in response to flg22, yet are impaired in MAMP-responsive actin remodeling and are more susceptible to infection than WT plants.  Please provide a more thorough discussion of these immunity phenotypes and their potential pleiotropic complications (e.g. pldb mutants were previously shown to affect SA/JA signaling; doi:10.1111/nph.12256). 

Line 350: The actin phenotype of the rbohd cpb-1 double mutant in Figure 6C is said to provide "genetic evidence that RBOHD-generated ROS negatively regulates CP function during the innate immune response." Given that the comparison used to support this statement is the cpb-1 single versus rbohd cpb-1 double mutant, conclusions cannot be drawn about the regulation of wild-type CP.  Instead, these data do indicate that ROS act upstream of CP for actin remodeling.
